# Gap Analysis of the Habitat Interface of Ticks and Wildlife in Mexico

**DOI:** 10.3390/pathogens10121541

**Published:** 2021-11-25

**Authors:** Carlos A. López González, Norma Hernández-Camacho, Gabriela Aguilar-Tipacamú, Salvador Zamora-Ledesma, Andrea M. Olvera-Ramírez, Robert W. Jones

**Affiliations:** 1Department of Ecology and Wildlife Diversity, Facultad de Ciencias Naturales, Universidad Autónoma de Querétaro, Santiago de Querétaro 76230, Mexico; carlos.lopez@uaq.mx (C.A.L.G.); norma.camacho@uaq.mx (N.H.-C.); salvador.zamora@uaq.mx (S.Z.-L.); rjones@uaq.mx (R.W.J.); 2Department of Animal Health and Environmental Microbiology, Facultad de Ciencias Naturales, School of Veterinary Medicine, Universidad Autónoma de Querétaro, Santiago de Querétaro 76230, Mexico; andrea.olvera@uaq.mx

**Keywords:** gap analysis, ticks, wildlife, tick borne diseases, zoonotic, Mexico

## Abstract

Mexico is a highly diverse country where ticks and tick-borne diseases (TBD) directly impact the health of humans and domestic and wild animals. Ticks of the genera *Rhipicephalus* spp., *Amblyomma* spp., and *Ixodes* spp. represent the most important species in terms of host parasitism and geographical distribution in the country, although information on other genera is either limited or null. In addition, information regarding the influence of global warming on the increase in tick populations is scarce or nonexistent, despite climate conditions being the most important factors that determine tick distribution. In order to aid in the management of ticks and the risks of TBD in humans and domestic animals in Mexico, an analysis was conducted of the gaps in information on ticks with the purpose of updating the available knowledge of these ectoparasites and adapting the existing diagnostic tools for potential distribution analysis of TBD in wildlife. These tools will help to determine the epidemiological role of wildlife in the human–domestic animal interface in anthropized environments in Mexico.

## 1. Mexican Ticks: How Many Species and Where to Find Them

Diversity and Distribution of Tick Species in Mexico

Mexico has 100 reported species of ticks, which corresponds to 11.2% of the known world diversity [1,2]. This percentage is consistent with the world diversity of other animal groups, such as that of Mexican birds and mammals, at 11.3% and 12.2%, respectively [3]. Two tick families are found in Mexico: *Argasidae*, or soft ticks, and Ixodidae, the hard ticks, the latter being the better studied. Of the *Argasidae*, there are 32 reported species within 5 families, which include *Argas* (6 species), *Antricola* (3), *Ornithodoros* (20), *Otobius* (2), and *Nothoaspis* (1). The Mexican Ixodidae have a reported total of 68 species within 5 genera that include *Ixodes* (26 species), *Amblyomma* (26), *Dermacentor* (10), *Haemaphysalis* (3), and *Rhipicephalus* (3) [1].

The genera *Amblyomma* and *Ixodes*, being the most species-rich genera, also have the most reported host species, and common species are well distributed. The main hosts of *Amblyomma* are mammals, followed by reptiles, birds, and amphibians, whereas *Ixodes* has only been recorded in association with birds and mammals [1]. Both genera are well distributed, with *Amblyomma* recorded in 30 of the 32 states of Mexico [4] and *Ixodes* in 26 of the 32 states. *Amblyomma mixtum* (*A. mixtum*, previously *A. cajennense*) has the greatest distribution recorded, in 30 states [4], whereas *Ixodes scapularis* (*I. scapularis*) has the widest distribution of its genus, with 13 reported states [5]. The family Argasidae in Mexico has been less studied than the Ixodidae. Of the 32 known species of argasids in the country, 20 are reported from the genus *Ornithodoros* [1]. These species are found associated with birds and mammals, often found in their sleeping and resting places. Several of the Mexican argasid species are found at roosting sites in caves with bats [2].

Despite the mega diversity of ticks recorded for the country and the study of parasitic fauna of terrestrial vertebrates in Mexico spanning more than 80 years, only about 20% of the total vertebrate species in the country (1145 species out of 5488 registered) have information available concerning their parasitic fauna [6]. Of these, mammals are one of the least-studied vertebrate groups, with only 121 recorded host species of ticks out of a total of 535 species. This is even more noteworthy given that mammals have the highest reported species richness of ticks when compared to other groups of vertebrates [7].

This information gap limits the study and management options of ticks and tick-borne diseases in Mexico. In particular, a better understanding of Mexico’s elevated diversity and ecological interactions of both ticks and their hosts has high importance in relation to human health and in future management decisions concerning the risk of zoonotic transmission with Mexican vertebrates [8]. Evidence of the gaps in information is apparent in the differences in the total recorded species richness of ticks among states. To date, the states with the highest tick species richness are Veracruz (26 species) and Oaxaca (23 species). However, the adjacent states of Puebla and Tabasco have fewer recorded species, most probably due to undersampling, as these states share similar climates and vegetation types (Figure 1).

The phenomenon of globalization further accentuates the need for increased information on the present species diversity and their distribution, as it has created a snowball effect on the colonization, expansion, relocation, and translocation of species and diseases, at intra- and inter-continental scales. These changes in and expansions of the distribution of parasites translate into negative effects for the native wildlife that coexist with us in the interface between natural and anthropogenic ecosystems.

Ticks have been associated with a variety of groups of wildlife, such as reptiles, birds, and mammals. The latter are the taxon with the largest number of documented terrestrial species. Likewise, there are few cases in which the presence of shared taxa among ticks has been documented [6], as most species of ticks show host specificity, reflecting co-evolution with their hosts. This process has manifested itself, above all, in endemic species and those with restricted distributions and unique biological and ecological characteristics [7]. Although Mexico has a great diversity of hosts and a significant number of tick records, the vast majority of these data are concentrated on domestic mammals and the wild fauna that occur in close proximity in human-dominated landscapes. However, little is known of the tick/animal associations in native terrestrial vertebrate populations. It is essential to increase studies on the richness and distribution of these arthropods associated with other groups of wild vertebrates, especially reptiles and birds. In particular, migratory birds are singularly important, because of their high mobility which increases dispersal of their parasites; this, combined with climate change, results in increased risks of zoonoses [7,8].

Tick species with a wide host spectrum are those that may pose a greater zoonotic risk for anthropogenic activities. Such is the case of the *Ixodes* and *Rhipicephalus* genera, due to their association with domestic artiodactyls or pet animals (such as cats or dogs). However, such risk increases with the number of contacts that humans have with ticks. Most of the records for these two genera occur in sites where land use has been transformed and converted to maximize agricultural and/or livestock production (Figure 2) [9].

In order to determine the distribution of the genera and species of ticks in Mexico, and to identify possible omissions, we obtained the records from the Global Biodiversity Information Facility (www.gbif.org accessed on 17 August 2021). This search yielded 8104 records relating to the different species and genera of ticks present in Mexico, which were used to document their association with vegetation types and productive management activities.

The distribution of ticks by type of vegetation in Mexico is shown in Figure 3; Figure 4. The environments with the greatest diversity of genera are xerophytic scrubland and tropical deciduous forests, both with eight genera. Grassland environments, dominated by native and introduced ungulates, do not have a great wealth of genera, and only *Otobius* is widely represented in this environment. The vegetation types with the lowest diversity of genera are montane cloud forests and wetland vegetation. It is extremely relevant that xerophytic scrubland presents such a high diversity, since these parasites have a codependency with particular humidity and soil type conditions [11]. The tick genus *Antricola* is exclusively distributed in tropical deciduous forest. Likewise, the genus *Haemaphysalis* is only associated with desert scrubland. Most of the records of the genus *Rhipicephalus*, namely, *Rhipicephalus sanguineus* (*R. sanguineus*), are in Xerophilous scrub (Figure 3). For all environments considered (n = 9), the majority of the genera are present in an intermediate distribution (4–6 environments), four are specialists (1–3), and two are generalists (7–9).

The genus with the highest number of records in the Global Biodiversity Information Facility (GBIF) is *Rhipicephalus*, mainly the species *R. sanguineus*, associated with the xerophilous scrub vegetation type (Figure 3). This species is traditionally associated with domestic, stray, and feral dogs [8,9]. The main types of vegetation where most genera of ticks are recorded are tropical deciduous forest, thorn forest, and Xerophilous scrub [9,12]. 

These data show the wide presence and distribution of these ectoparasites throughout the country; this has a direct effect on the presence of pathogens associated with these organisms, which will be discussed in the next section.

## 2. The Potential of Ticks as Vectors of Pathogens in a Megadiverse Country

Vector-borne diseases represent approximately 17% of infectious diseases and cause more than 700,000 deaths annually worldwide [13]. In Mexico, the occurrence of vector-borne diseases such as Rocky Mountain spotted fever (RMSF) and Lyme borreliosis (LB) has been reported in news outlets. In northern Mexico, a high incidence of RMSF has been reported, with mortality reaching 32%, making it the country with the highest reported RMSF incidence [14,15]. In the case of LB, Colunga-Salas et al. [16] reported 98.7% (393/398) of known LB cases, of which 35.6% (140/393) of cases were identified with *Borrelia afzelii* (*B. afzelii*), *B. garinii*, and *Borrelia burgdorferi* s.s.

There has been increased interest and research directed at diseases in wildlife in order to understand their role as reservoirs of diseases that impact domestic and human animal health. Vector-borne diseases associated with wildlife coupled with climate change favor the re-emergence of diseases and the possible emergence of new ones. Reports of emerging zoonotic diseases have skyrocketed in recent years in response to encroachment, fragmentation, and loss of habitat. These disturbances affect wildlife population dynamics, by modifying distributions or increasing population densities—conditions favorable for epidemic outbreaks and epizootics. In order to understand these interactions, it is necessary to study the ecology, evolution, and biogeography of the parasite–host system [17,18].

Ticks are one of the most efficient arthropods for the transport of pathogens that can cause serious illness. Attempts at eradication through the use of chemical and non-chemical compounds has led to species resistance, in addition to contamination of the environment [19]. Combined with the problems of resistance, the epidemiology of tick-borne diseases is affected by anthropogenic and natural actions such as changes in land use, climate, and the introduction of pathogens to disease-free areas. The emergence of a new disease is not a simple process, nor is it easily established. First, the pathogen must be introduced into a new population, followed by survival and dissemination; finally, under optimal conditions, the disease becomes established. These are complex systems that are subject to changes in ecological processes that influence the biology of the vector and, therefore, the epidemiology of the pathogen [17].

Tick-borne diseases such as borreliosis, ehrlichiosis, rickettsiosis, and anaplasmosis are considered emerging and re-emerging diseases of importance for Public Health [20]. However, there are viral zoonotic pathogens and other bacterial types transmitted by ticks for which the epidemiological role that wildlife play in the maintenance of these diseases in the ecosystem is unknown. In order to understand the epidemiology of tick-borne diseases, the following information is needed: pathogen distribution patterns, evolutionary history between pathogen and host, susceptible wildlife or domestic populations, probable risk areas, environmental factors, and determinants of disease dynamics as influenced by spatio-temporal patterns [17,18]. These factors will help to understand the emergence and re-emergence of zoonotic pathogens, as any change in the aforementioned factors can modify the dynamic balance of wild reservoirs and their associated pathogens [21].

Several tick-borne diseases that have been reported in Mexico are associated with wildlife and are of importance to public health. The first of these is the Rocky Mountain spotted fever, caused by the bacteria *Rickettsia rickettsii* (*R. rickettsia*) and transmitted by *Dermacentor*, *Rhipicephalus*, and *Amblyomma*, detected in dogs and humans [22,23]. Anaplasmosis and ehrlichiosis are infectious bacterial diseases that can be transmitted to vertebrates by infected ticks. *Anaplasma phagocytophilum* (*A. phagocytophilum*) has been reported for domestic species in Mexico (sheep, goats, cows, horses, dogs, cats, roe deer, reindeer) and humans [24,25]. *Erlichia* spp. is transmitted by *R. sanguineus* and *A. mixtum*. White-tailed deer (*Odocoileus virginianus*) are an efficient reservoir for *E. chaffeensis* [26]. In addition, several highly important tick-borne diseases such as babesiosis may have wildlife reservoirs; for example, Nilgai antelopes (*Boselaphus tragocamelus*) can be positive for *Babesia bigemina* (*B. bigemina*) and *Babesia bovis* (*B. bovis*). Although the epidemiological role of this species has not been determined, the possibility cannot be discarded that it may serve as a potential ecological reservoir of these diseases in domestic animals [27]. The role of native Mexican fauna that may serve as reservoirs of tick-borne diseases requires further study.

In the studies carried out in Mexico concerning pathogen–vector–wildlife interaction described in Table 1, it is observed that most of these used the conventional PCR test to detect the pathogens. However, some authors are already beginning to use qPCR for the detection of multiple pathogens, using a single sample, which is advantageous due to the greater difficulty in obtaining biological samples from wild fauna. In addition, handling of wild animals can cause stress and death, so it is important to have rapid diagnostic tests with greater sensitivity and high specificity.

Additionally, most studies of pathogens in wild fauna are focused on the detection of bacterial diseases, mainly borreliosis, rickettsiosis, and ehrlichiosis, and parasitic diseases such as babesiosis and anaplasmosis. Few studies focus on the detection of viral agents, which is an area of opportunity because many zoonotic diseases are of the viral type. Information on the agents that may be present in wild fauna in Mexico is lacking. There is also a notable dichotomy in many studies between those that search for pathogens in the host and those that do so only in the vector. Few studies look for the presence of the vector and pathogen in the wild host. In some studies, the pathogen has been detected but the vector has not been found; therefore, the epidemiological role of the vector in the transmission of the biological agent cannot be explained. The absence of ticks in the wild fauna that has been sampled could be associated with climatic factors that affect the survival of free-living ticks, in addition to the abundance according to the season of the year in which the sampling was carried out and, therefore, its ability to transmit the pathogen. Another important factor is behavioral habits, such as in rodents, which constantly clean their bodies, preventing ticks from attaching [28,29].

## 3. Diagnostics for the Detection of Tick-Borne Pathogens

An important part of detecting the biological agents transmitted by ticks is the proper use of diagnostic tests. These are designed to determine whether wild animals have circulating antibodies or if the animal is subject to an infection. The diagnostic tests available and their characteristics for the detection of tick-borne pathogens are detailed below.

### 3.1. Serological Diagnostics

The most frequently used serological tests for the diagnosis of tick-borne diseases include enzyme-linked immunosorbent assay (ELISA), indirect immunofluorescence assay (IFI), and Western blot. However, it is important to note that the sensitivity and specificity of these techniques may vary due to different factors, such as sample collection time, antibody kinetics, and assay methodology, among others [41]. Despite these variations, they have been widely used in the serological detection of pathogenic microorganisms transmitted by ticks in wildlife. For example, Salinas-Meléndez et al. [42] used the indirect immunofluorescence test (IFI) to detect antibodies against *B. burgdorferi*, in 850 blood samples obtained from dogs in Monterrey, Mexico, of which 16% (136) of these dogs were positive. Panti-May et al. [36] detected 80.9% (17/21) antibodies against *R. rickettsii* and *R. tiphy* by IFI in rodents. Romero-Salas [43] detected antibodies against *B. bovis* and *B. bigemina* in water buffalo, at 71.4% (110/154) and 85% (125/154), respectively, using IFI. In contrast, Aguilar-Tipacamú et al. [14] detected 100% (92/92) antibodies against *A. paghocytophilum* and *E. canis* by indirect immunofluorescence (IFI) in wild mice in Querétaro, Mexico. New techniques have been developed to improve sensitivity and to be able to detect more diseases with fewer trials. Tokarz et al. [44] developed the first serological diagnostic method for multiple tick-borne diseases in humans, known as TBD-serochip, which identifies *A. phagocytophilum*, *Babesia microti*, *B. burgdorferi*, *B. miyamotoi*, *E. chaffeensis*, *R. rickettsii*, Heartland virus, and Powassan virus. This method is based on microarrays, in which the matrix has up to 3 million linear 12-mer peptides that can be divided into 12 subarrays, each of which contains approximately 170,000 12-mer peptides.

### 3.2. Molecular Diagnostics

Molecular diagnostic techniques have been widely used to identify tick-borne pathogens. PCR has been widely recommended as it can confirm an infection or detect multiple microorganisms in a single assay. Romero-Salas et al. [43] used end-point PCR and nested PCR with oligonucleotides that they themselves designed to detect the apocytochrome b genes of *B. bovis* and *B. bigemina* in the blood of water buffaloes and bovines in Veracruz, Mexico. This method has greater sensitivity with nested PCR. Solis-Hernandez et al. [30] identified *B. burgdorferi* in synanthropic rodents in Yucatán, México, using specific oligonucleotides described by Jaulhac et al. [45] to identify flagellin B (FlaB) genes and external membrane lipoproteins (OspC/p66). In the case of the real-time PCR technique (qPCR), it allows the detection of a single sequence (singleplex) [46] or multiple sequences (multiplex) [47]. However, current qPCR platforms limit the number of probes detected with a given instrument, thus limiting the number of pathogens detected in a single test. Most of the widely used qPCR platforms are limited to 4–5 fluorophores in a single reaction without interfering with pathogen detection [48]. Molecular assays to detect nine or more pathogens have recently been described using high-definition PCR, which is a multiple molecular assay that detects nine individual pathogens or a group of pathogens associated with tick-borne diseases [49]. Modarelli et al. [48] developed an assay that detects 11 pathogens (TickPath Layerplex qPCR), in which the qPCR method is used and can detect and characterize 11 pathogens that cause tick-borne disease in domestic dogs simultaneously, using the same melting temperature but with different fluorogenic probes. This was demonstrated to be compatible with qPCR thermal cyclers used in molecular diagnostics. Yu et al. [31] conducted a study in Texas to detect tick-borne pathogens in blood, using the methodology described by Modarelli et al. [48], detecting *B. vogeli* and *B. tuncatae* in coyote (*Canis latrans*), as well as *T. cervi*, *E. chaffensis*, and *A. platys* in white-tailed deer (*Odocoileus virginianus*). Ojeda-Chi et al. [32] used the same TickPath layerplex qPCR essay, first to identify pathogens of the Anaplasmataceae and Rickettsiaceae family in the liver and spleen tissue of white-tailed deer (*Odocoileus virginianus*) and red temazate (*Mazama temama*), as well as ticks obtained from these. They later used a nested PCR technique to identify the fragments of the 16S and ompB rRNA genes and thus detected *Ehrlichia*, *Anaplasma*, and *Rickettsia*, which were later confirmed by sequencing.

## 4. Chronicle of an Announced Zoonosis: The Effect of Global Climate Change on Ticks and Their Associated Pathogens

The existence of ticks in a given locality requires two factors: the presence of its hosts and the appropriate environmental conditions. These factors allow the organism to maintain a dynamic ecological interaction with multiple microorganisms ranging from endosymbionts to pathogens [38,50]. However, human activities have added additional factors that were not present during the co-evolution process of the parasite–host relationship. Most notably, global climate change has severely affected the stability and distribution of one of the oldest lifestyles on the planet [11]. In the case of ticks, their host dependence has a direct influence on their distribution and ability to find members of the same species to reproduce and maintain their genetic diversity. This dependency is also influenced by their feeding habits, including the use or not of burrows, and their variable tolerance to desiccation [51]. This last point is of utmost importance, considering that ticks and their role as vectors of zoonotic diseases are strongly dependent on the environmental conditions of the regions where they are found; current anthropogenic modifications of the environment indicate that the vector ecology of ticks is under considerable transformation [52,53].

Currently, the severity of the impact that climate change has on the ecosystems of Mexico on a large scale is not well understood. An analysis by Cuervo-Robayo et al. [54] compiled and processed weather station data of climate conditions in Mexico during the last hundred years (1910–2009). Although these data have limitations given the length of time involved and inevitable variation in the number of working climate stations, the data provide an overview of what to expect in the future. This analysis shows that during the last century, the national average temperature has increased by 0.2 °C. However, this has not occurred uniformly throughout the country, since there may be variations between the different regions of Mexico. With respect to precipitation, it also indicated a temporary increase during the decades from 1940 to 1970 but showed a constant decrease from the end of that period to the present. Therefore, the characteristics of the latitude and orography of the country, which have a direct relationship with the climatic conditions of a particular region, together with the role of global climate change, will play a very important role in the range of distribution, survival, abundance, and vector role that ticks will have in Mexico.

Recent studies focused on the effect of climate change on the distribution of ticks in Africa and Europe show a constant increase in the presence of diseases transmitted by ticks in areas where they had previously not been recorded [11,53,55]. This occurred particularly in those sites where humidity and temperature have increased in recent years, as these parasites are highly sensitive to changes in these two factors. Therefore, a similar scenario may be expected in the case of Mexico, considering Cuervo-Robayo et al. [54] and [56], with an increase in tick distribution [57]. Consequently, an increase in the risk of transmission of pathogens to wildlife, domestic species, and humans is expected in the coming years. This disturbing consensus is manifested in the latest IPCC report, published in early August 2021 [58], concluding that it is no longer possible to reverse the effect of global climate change and environmental modifications are already in process, so we must expect important ecological changes in the next decade.

Titcomb et al. [59] demonstrated that climate change and the loss of wildlife diversity affect the prevalence of zoonotic diseases. The exclusion of wildlife affects the abundance of ticks, and this effect varies over time. In Mexico, studies have not been carried out to explore the effect of habitat fragmentation, loss of wildlife diversity, and climate change on the presence of tick-borne diseases; this is an important area of opportunity to understand the future dynamics of tick-borne diseases in the country.

## 5. Conclusions

What is the future in Mexico regarding ticks? It is clear that there is a lack of important information on ticks associated with wildlife, mainly in birds and mammals. The available information is highly oriented towards domestic animals parasitized by tick species in systems with a wide range of hosts. Despite having a long history of parasitological studies in Mexico and being a megadiverse country, it is disconcerting that the inventories of parasites of wildlife are wholly inadequate, especially regarding those with zoonotic potential. The current situation of the COVID-19 pandemic must be considered as a red flag of the consequences of ignoring the zoonosis to come—a proven mistake; these results could be associated with wildlife–domestic animal–human interaction in fragmented areas, where the aforementioned types of vegetation are present. Although many of the tick species are host specific, those that are more abundant in anthropized environments parasitize various taxa, which makes them dangerous when in contact with humans. Therefore, it is essential to start work where the host, the agent, and the reservoir are detected [60]. Also needed are in situ experimental studies, using wild animals as epidemiological models to assess their susceptibility to tick-transmitted disease and to understand their role in the transmission and maintenance of these diseases.

Regarding the distribution of ticks, the states with the highest livestock activity are those that present a greater richness of tick species, as is the case of Veracruz and Oaxaca, which also coincides with mammalian species richness. However, this does not mean that there is an individual and unique tick species for each wild mammal species, but a high possibility of transmission of these ectoparasites between different host species populations. This is especially important for artiodactyls, which are considered of greatest importance among mammals for the evolution of ticks [7,61]. The three genera with the highest representation in mammals in Mexico are *Rhipicephalus*, *Amblyomma*, and *Ixodes*, the first being the one with the highest number of records in the country according to the GBIF. These same three genera of ticks are distributed in areas of xeric scrublands, while most of the genera *Rhipicephalus*, *Carios*, *Dermacentor*, *Ixodes*, *Amblyomma*, and *Antricola* present in Mexico are recorded in tropical deciduous forest. An area of opportunity in wildlife research is to increase the records on the species richness of ticks using the DNA barcode method, given that it is now a fast and reliable method compared to the traditionally used morphology-based identification [62].

## Figures and Tables

**Figure 1 pathogens-10-01541-f001:**
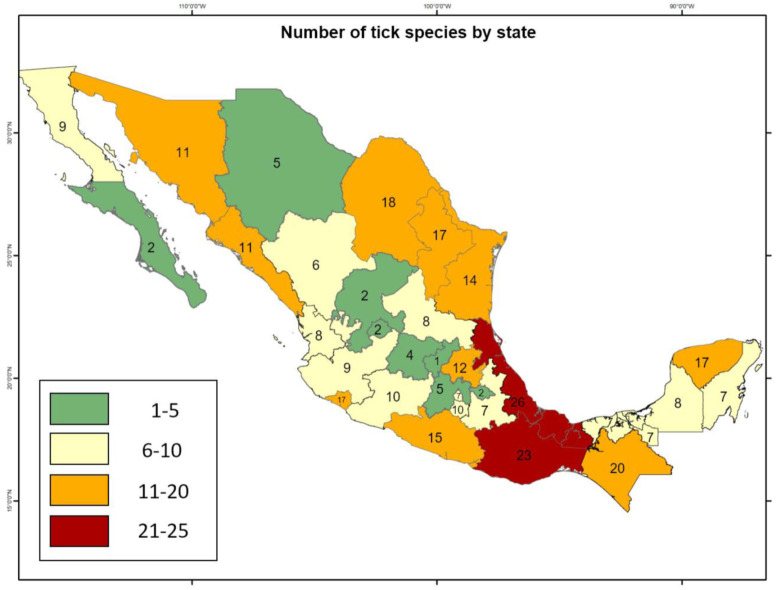
Tick species richness by state in Mexico (map created using the program ArcMap 10.3 and records from [9]).

**Figure 2 pathogens-10-01541-f002:**
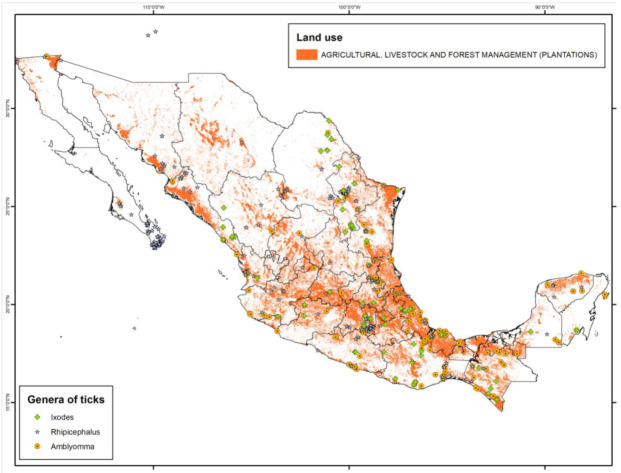
Tick genera associated with agriculture, livestock, and forestry land use throughout Mexico (map created using the program ArcMap 10.3 and records from [10]).

**Figure 3 pathogens-10-01541-f003:**
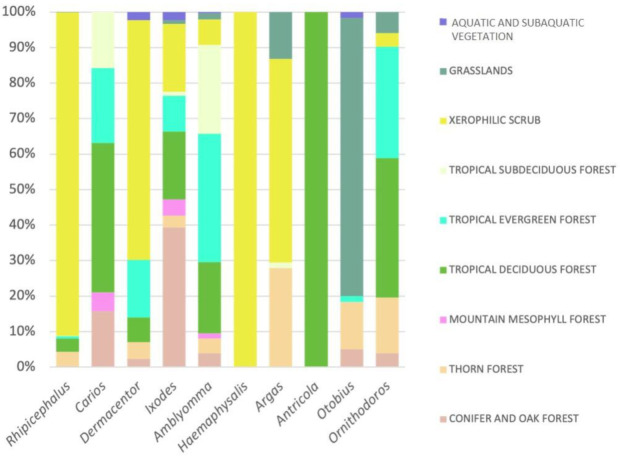
Tick genus distribution percentages (%) for the types of vegetation in Mexico (graph created using the program Microsoft Excel 365 and records from [10]).

**Figure 4 pathogens-10-01541-f004:**
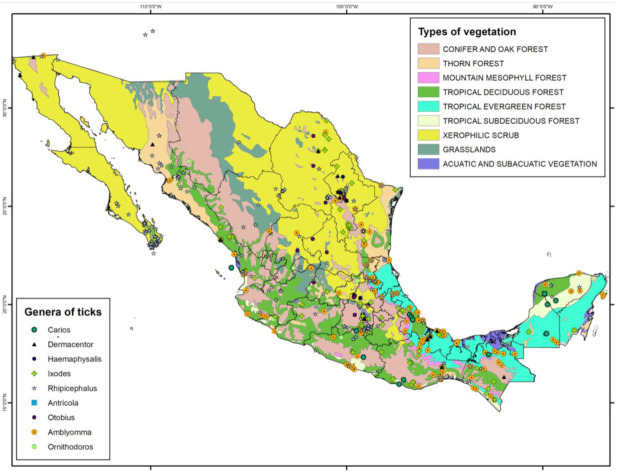
Distribution of tick genera in accordance with types of vegetation (map created using the program ArcMap 10.3 and records from [12]).

**Table 1 pathogens-10-01541-t001:** Tick-borne pathogens surveyed by molecular techniques in Mexican wildlife.

Pathogen	Pathogen Frequency (%)	Type of Sample	Ticks	Stage	Molecular Detection Technique	Host	State	Reference
*Borrelia burgdorferi* (*B. burgdorferi*) sensu lato	40/94 (42.5)	Bladder/ear	-	-	PCR	House mouse (*Mus musculus*)	Yucatan	[30]
*B. burgdorferi* sensu lato	5/29 (17.5)	Bladder/ear	-	-	PCR	Rats (*Rattus rattus*)		
*A. phagocytophilum* *Ehrlichia canis* (*E. canis*)	48/54 (88.9) 53/59 (89.8)	Serum /blood	-	-	PCR	Mexican deer mouse (*Peromyscus mexicanus*)	Querétaro	[14]
*A. phagocytophilum* *E. canis*	6/54 (11.1) 6/59 (10.1)	Serum /blood		-		Rat (*Rattus rattus*)		
*Babesia vogeli* (*B. vogeli*)	11/22 (9)	Blood	-	-	TickPath Layerplex qPCR	Coyote (*Canis latrans*)	Texas (USA-Mexico border)	[31]
*Babesia turicate* *Ehrlichia chaffensis* (*E. chaffensis*) *Theileria cervi* (*T. cervi*) *Anaplasma platys* (*A. platys*)	1/122 (0.8) 4/245 (1.6) 18/245 (7.3) 1/245 (0.4)	Blood	-	-	TickPath Layerplex qPCR	White-tailed deer (*Odocoileus virginianus*)		
*Anaplasma odocoilei* (*A. odocoilei*), *A. phagocytophilum and E. chaffensis*	5/25 (20)	Spleen/liver	*A. mixtum*, *Amblyomma parvum* *A. cf. oblongoguttatum*, *Ixodes affinis*, *R. microplus and R. sanguineus* sensu lato, *and Haemaphysalis juxtakochi.*	-		White-tailed deer (*Odocoileus virginianus yucatensis*)		[32]
*A. odocoilei*, *A. phagocytophilum* *and E. chaffensis*	2/4 (50)				TickPath Layerplex qPCR, nested PCR	Mazama deer (*Mazama temama*)	Yucatan	
*B. burgdorferi*	8/25 (29)	-	*I. scapularis*	-		Eastern cottontail (*Sylvilagus floridanus*)	Nuevo Léon	[33]
*B. burdorferi*	1/1 (100)	-	*I. scapularis*		PCR	Jaguar (*Panthera onca*)	Tamaulipas	
*B. burdorferi*	1/6 (0.16)	-	*I. scapularis*			Painted spiny pocket mice (*Liomys pictus*)	Nuevo León	
*R. rickettsii*	3/60 (5)	-	*Dermacentor variabilis* (*D. variabilis*)		PCR	Bobcoat (*Lynx rufus*)	Tamaulipas	[34]
*Rickettsia bellii*	1/37 (2.7)	-	*Haemaphysalis leporispalustris*	Adults	PCR	Rabbits (*Lepus* sp.)	Hidalgo	[35]
*Rickettsia felis*	10/23 (43.5) 4/7 (57.1)	Spleen	-	-	PCR	Rodents (*Mus musculus*, *Heteromys gaumeri*, *Sigmodon hispidus*, *Olgorizomys sp. Peromyscus yucatanis*) Virginiana opposum (*Didelphis virginiana*)	Yucatan	[36]
*Rickettsia* sp.	2/16 (12.5)	-	*R. sanguineus* s.1.	Adults	PCR	Coyote (*Canis latrans*)	Chihuahua	[37]
*Borrelia* sp.	1/5 (20) 2/31 (6.4)	-	*Ixodes kingi* *R. sanguineus* s.1.			Kit fox (*Vulpes macrotis*) Free roaming dogs (*Canis lupus familiaris*)		
*Rickettsia parkeri*	6/22 (27.3)	-	*A. ovale*	Adults	PCR	Owned ad Free roaming dogs (*Canis lupus familiaris*)	Veracruz	[38]
*Coxiella burnetti*, *A*. *phagocytophilum*, *Neorerlichia sp.*		Tick pool	*Ornithodoros turicata*		Massive sequencing	Bolson tortoise (*Gopherus flavomarginatus*)	Chihuahua, Coahuila, Durango	[39]
*Rickettsia monacensis* (*R*. *monacensis*)	3/16 (19.7) 2/14 (14.3)	-	*Amblyoma dissimile*	Nymphs Adults	PCR	Common boa (*Boa* *constrictor*)	Veracruz	[38]
*R. monacensis* *R. monacensis*	1/9 (11.1) 1/2 (50)	-	*Amblyoma dissimile* *A. mixtum*	Adults Adults		Green iguana (*Iguana iguana*)		
*R. monacensis*	2/15 (13.3) 1/16 (6.3)	-	*Amblyoma dissimile*	Larvae Nymphs		Marine toad (*Rinella marina*)	Guerrero	
*R. monacensis*	3/25 (12)	-	*Amblyoma dissimile*	Larvae		Marine toad (*Rinella marina*)	Tabasco	
*Borrelia* sp.	5/60 (8.3)	-	*Amblyoma dissimile*	Adults	PCR	Mesoamerican canean toad (*Rhinella horribilis*)	Veracruz	[40]

## Data Availability

Data used for the construction of figures is available upon request from the authors.

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
