# Peer review of "Gap Analysis of the Habitat Interface of Ticks and Wildlife in Mexico"

_pathogens, 2021, doi:10.3390/pathogens10121541_

Round 1

Reviewer 1 Report

Interesting research overview, well documented by literature. Is getting tick-borne diseases in Mexico a big medical problem? How many people are diagnosed annually? Brief information on this could be included in the chapter "The potential of ticks as vectors of pathogens in a megadiverse country".

Author Response

Response: Thanks for the comment the manuscript was reviewed, analyzed and writing by native speaker (Robert W.Jones).

Reviewer 2 Report

The authors attempt to close the gap pertaining to the current information on ticks. The information in this review is relevant and the authors present their results appropriately. The manuscript, however, is need of English corrections from a native speaker.

Author Response

REVIEWER 2

We appreciate again the effort and time put on this revision by the reviewers. Below are the answers to each of your questions and comments.

  1. Line 45. Delete “Say”

Response: Thanks for the comment, Line 45: We delete “Say”

  1. Line 55: Delete “s” in vertebrate

Response: Thanks for the comment, Line 55: We delete “s” in vertebrates

  1. Line 59. Delete “and”

Response: Thanks for the comment, Line 59: we delete “and”

  1. Line 62: add a “in” in concerning and separate “therisk”

Response: Thanks for the comment, Line 62 we add concerning “in” and separate “the risk”

  1. Line 63: Delete “Is”

Response: Thanks for the comment, Line 63: we delete “is”

  1. Line 73: Delete “e”

Response: Thanks for the comment, Line 73: We delete “e” parasites

  1. Line 81: Delete “most”

Response: Thanks for the comment, Line 81: We delete “most”

  1. Line 92: Add reference in “Zoonoses”.

Response: Thanks for the comment, Line 92: We add reference (7,8).

  1. Line 135: [13], there changed “There”

Response: Thanks for the comment, Line 143: we change by There

  1. Line 138: Delete “makes”

Response: Thanks for the comment, Line 138 we delete “makes”

  1. Line 147: harmful what???

Response: Thanks for the comment, Line 155: we add “addition to contamination of the environment [19]”

  1. Line 149: Delete “t”

Response: Thanks for the comment, Line 158 we delete “t”

  1. Line 151: Delete “a”

Response: Thanks for the comment, Line 160 delete “a”

  1. Line 152: put “survival” first and followed by “dissemination”

Response: Thanks for the comment, Line 160: we change “followed by survival and dissemination”

  1. Line 156: names of diseases are usually not written with capital letters Borreliosis…….

Response: Thanks for the comment, Line 164 we change by borreliosis, ehrlichiosis, rickettsiosis and anaplasmosis……..

  1. Line 161: Delete “i:”

Response: Thanks for the comment, Line 169: we add “is needed”:

  1. Line 167: Delete “that” are associated

Response: Thanks for the comment, Line 175: we delete “that”

  1. Line 168: Delete one “the”

Response: Thanks for the comment, Line 176: we delete “the”

  1. Line 171: Change “Ehrlichiosis” by “ehrlichiosis”

Response: Thanks for the comment, Line 179 we change by “ehrlichiosis”

  1. Line 177: Delete “.”

Response: Thanks for the comment, Line 185: Delete “.” and add “,”  for example

  1. Line 185: something is missing in “capture”.

Response: Thanks for the comment, Line 194-197 we change by “using a single sample, which is advantageeous due to the greater difficulty in obtaining biological samples from wild fauna. In addition, handling of wild animals can cause stress and death, so it is important to have rapid diagnostic tests with greater sensitivity and high specificity

  1. Line 189: Explained, what does “them” refer to. Here causality is not directly understood, rewording needed

Response: Thanks for the comment, Line 198: we change by “most studies of pathogens in wild fauna are focused on the detection·

  1. Line 193: Add “I” in t

Response: Thanks for the comment we changed Line 203-205 by “There is also a notable dichotomy in many studies between those that search for pathogens in the host and those only in the vector”

  1. Line 203: Table 1 change n/n (%)

Response: Thanks for the comment we changed n/n (%) of the table 1 the column Pathogen frequency

  1. Line 220: Panti-May …. by IFI in wich 80-90 % especimens/hosts? IFI respectively ….

Response: Thanks for the comment we changed Line 237-240  by “detected 80.9% (17/21) antibodies against Rickettsia rickettsii and R. tiphy by IFI in rodents. Romero-Salas [43] detected antibodies against B. bovis and B. bigemina in water buffalo, 71.4% (110/154) and 85% (125/154) respectively, using IF”.

  1. Line 223: Delete “Where”

Response: Thanks for the comment we delete Line 240. “where”

  1. Line 226: Delete “to identify them” ….

Response: Thanks for the comment we delete Line 244 “to identify them”

  1. Line 252. Detection and characterización “To be compatible”

Response:  Thanks for the comment we add “detection and characterization to be compatible to 11 pathogens that cause tick-borne disease in domestic dogs, demonstrating to be compatible with qPCR thermal cyclers used in molecular diagnostics. Yu et al. [31]

  1. Line 264-265: Chronicle of a preordained? zoonosis: the effect of global climate change on ticks and their associated pathogens

Response: Thanks for the comment we changed Line 281-282 by “Chronicle of an announced zoonosis: the effect of global climate change on ticks and their associated pathogens”

  1. Line 284: Delete after May “Be”

    Response: Thanks for the comment we changed Line 302-304 by Although     these data have limitations given the length of time involved and inevitable   variation in the number of working climate stations, the data

  1. 31. Line 296: Add “and”

Response: Thanks for the comment we add “Europe and show”

  1. Line 302: “Not clear what is ment by tick species”

Response: Thanks for the comment, Line 309 we analized and deleted “In Mexico, studies have not been carried out that explore the effect of habitat fragmentation, loss of wildlife diversity and climate change on the presence of tick-borne diseases; an important area of opportunity to understand the future dynamics of tick-borne diseases in the country”

  1. Line 330: Delete as

Reviewer 3 Report

 In this review article, the authors provide updated information on the tick populations and their association with different ecosystems in Mexico and review literature on the association between climate change and (re)appearance of zoonoses as well as provide some guidance/proposals on how to conduct future studies on the epidemiological aspects of zoonoses, particularly the aspect of vector-borne pathogens distribution.

I propose additional English editing of the manuscript.  

Author Response

Response: Thanks for the comment, Line 136-142: we add “In Mexico, the occurrence of vector-borne diseases such as Rocky Mountain spotted fever (RMSF) and Lyme borreliosis (LB) has been reported in news outlets. In northern Mexico, a high incidence of RMSF has been reported, with mortality reaching 32%, making it the country with the highest reported RMSF incidence [14,15]. In the case of LB, Colunga-Salas et al., [16] reported 98.7% of (393/398) of known LB cases, of which 35.6% (140/393) of cases were identified with B. afeliisB. garrinii and Bburdorferi s.s.”

Round 2

Reviewer 2 Report

English are fine.

Author Response

We reviewed the manuscript for grammatical and orthographic errors, one of the co-authors is a native English speaker and carried out a thorough review of the manuscript.